



# Combination Analysis of Multi-Wavelength, Multi-Parameter Radar Measurements for Snowfall

Mariko Oue[1], Pavlos Kollias[1,2], Sergey Y. Matrosov[3], Alessandro Battaglia[4,5], and Alexander V. Ryzhkov[6]

1. Stony Brook University, Stony Brook NY, USA
2. Brookhaven National Laboratory, Upton, NY, USA
3. Cooperative Institute in Research in Environmental Sciences, University of Colorado and NOAA Physical Sciences Laboratory, Boulder, CO, USA
4. Politecnico di Torino, Turin, Italy
5. University of Leicester, Leicester, UK
6. NSSL, University of Oklahoma, Norman, OK, USA

## Abstract

Radar dual wavelength ratio (DWR) measurements from the Stony Brook Radar Observatory Ka-band Scanning Polarimetric Radar (KASPR, 35 GHz), a profiling W-band (94 GHz) and a next generation K-band (24-GHz) Micro Rain Radar (MRRPro) were exploited for ice particle identification using triple frequency approaches. The results indicated that two of the radar frequencies (K- and Ka-band) are not sufficiently separated, thus, the triple radar frequency
approaches had limited success. On the other hand, a joint analysis of DWR, mean vertical Doppler velocity (MDV), and polarimetric radar variables indicated potential in identifying ice particle types and distinguishing among different ice growth processes and even in revealing additional microphysical details.

We investigated all DWR pairs in conjunction with MDV from the KASPR profiling measurements and differential reflectivity ($Z_{DR}$) and specific differential phase ($K_{DP}$) from the KASPR quasi-vertical profiles. The DWR-versus-MDV diagrams coupled with the polarimetric observables exhibited distinct separations of particle populations attributed to different rime degrees and particle growth processes. In fallstreaks, the 35–94 GHz DWR pair increased with the
magnitude of MDV corresponding to the scattering calculations for aggregates with lower degrees of riming. The DWR values further increased at lower altitudes while $Z_{DR}$ slightly decreased, indicating further aggregation. Particle populations with higher rime degrees had a similar increase of DWR, but the 1–1.5 m s$^{-1}$ larger magnitude of MDV and rapid decreases in $K_{DP}$ and $Z_{DR}$. The analysis also depicted the early stage of riming where $Z_{DR}$ increased with the MDV magnitude
collocated with small increases of DWR. This approach will improve quantitative estimations of snow amount and microphysical quantities such as rime mass fraction.

## 1. Introduction

Millimeter-wavelength (i.e., operating at 35- and 94-GHz) radars have been widely used for the study of liquid and ice precipitation clouds, utilizing their high sensitivity to smaller particles due to Rayleigh scattering and excellent spatiotemporal resolution (Kollias et al., 2007). Cloud radars at 35 GHz and 94 GHz have been routinely operated at surface-based observatories the last two decades (e.g., the European Union CloudNet project and the U.S. Atmospheric Radiation
Measurement (ARM) facility, Illingworth et al., 2015; Stokes and Schwartz, 1994; Mather and Voyles, 2013; Kollias et al., 2014; 2016) and from a variety of ship-based (e.g., Lewis et al., 2012) and airborne platforms (e.g., Battaglia et al., 2016; Tridon et al., 2019). Millimeter wavelength



radar are particularly suitable for the study of hydrometeors properties (mass, size) using the presence of non-Rayleigh scattering signals and their higher sensitivity to attenuation. For example, the dual-wavelength ratio (DWR), the ratio of the longer-wavelength reflectivity to the shorter-wavelength reflectivity, is affected by the differential scattering and/or attenuation and depends on the particle size, type, orientation, rime fraction, and radar beam path. DWRs have been used in multi-wavelength radar measurements for microphysical retrievals such as estimations of liquid water content (e.g., Hogan et al., 2005; Huang et al., 2009; Tridon et al., 2013; Zhu et al., 2019) and ice water content (IWC, e.g., Matrosov, 1998) and identification of particle types (e.g., Kneifel et al., 2015; Leinonen and Moisseev, 2015; Moisseev et al., 2015; Sinclair et al., 2016; Matrosov et al., 2019).

Kneifel et al. (2015) illustrated the effectiveness of DWRs to identify ice crystals, aggregates, and rimed particles, when considering well separated triple radar frequencies (i.e., X, Ka, and W bands) so that each frequency experiences different scattering regimes. The DWR of X-band to Ka-band reflectivities ($DWR_{XKa}$) versus DWR of Ka-band to W-band reflectivities ($DWR_{KaW}$) diagrams indicated different dependencies on particle type and size. Those curves were in good agreement with the observed particle types (Kneifel et al., 2016). The triple-frequency capabilities have been used for different frequencies such as S, X, and W bands and Ku, Ka, and W bands (e.g., Leinonen and Moisseev, 2015; Mason et al., 2019), or even shorter-wavelength radars (e.g., Ka, W, and G bands, Lamer et al. 2020). While the triple-frequency approach is a powerful technique for microphysics research, it requires accurate calibration of the radars, reliable attenuation correction, careful beam matching, and sufficiently high sensitivities at all frequencies. These conditions are satisfied only in a hand-full of surface observatories.

Another limitation of the triple frequency measurements for ice particle identification is that the triple frequencies should be well-separated from each other so magnitudes of non-Rayleigh scattering is different for various DWR pairs and the curves representing a particle type in the DWR correspondence diagram can be distinguished. If the frequencies are too close, then the DWR trends corresponding to different hydrometeor types may not be sufficiently separated from each other. For instance, Ka-band (around 35 GHz) frequency and K-band (around 24 GHz) frequency, which has been employed by a widely-used, low-power, low-cost, high-quality precipitation profiler, micro rain radar (MRR, e.g., Peters et al., 2002), are rather close, producing similar trends when coupled with a third frequency as shown in Fig. 1. Figure 1a is a $DWR_{KKa}$ versus $DWR_{KW}$ diagram from the scattering calculations (detailed descriptions of the scattering calculations are available in Appendix). Similarly, Fig. 1c is a $DWR_{XKa}$ versus $DWR_{KaW}$ diagram. These diagrams show that the two-DWR space from the three frequency radars exhibits a dependency on ice particle types, specifically size and rime fraction. However, considering modeling uncertainties and measurement noise, it would be hard to discern the particle types in the K-Ka-W DWR space, while the X-Ka-W DWR space has larger dynamic ranges likely enough to discern the particle types as presented in the previous studies. This is, in part, due to the fact that the K-band frequency (~24 GHz) is not sufficiently separated from the Ka-band frequency (~35 GHz).

It has been shown (e.g., Matrosov et al., 2019) that the DWR also depends on particle shapes (i.e., aspect ratios defined as the ratio of particle minor and major dimensions). For particles preferentially oriented with their major dimensions in the horizontal plane, the DWR dependence on particle shapes is usually strongest for vertically pointing radar measurements and relatively weak for slant radar viewing (Matrosov, 2021). The impacts of particle shape on the two DWR


pair's diagram (Fig. 1), however, is much smaller than that on individual DWRs for a given frequency pair. To illustrate this point, Figs. 1b and 1d show $DWR_{KKa}$–$DWR_{KW}$ and $DWR_{XKa}$–$DWR_{KaW}$ correspondences, respectively, for a "soft" spheroidal particle model with aspect ratios 0.3 and 0.8. A much weaker particle shape influence on this the $DWR_{KKa}$–$DWR_{KW}$ field (compared to individual DWRs) is explained, in part, by the fact that both $DWR_{KKa}$ and $DWR_{KW}$ increase/decrease as particle become more/less spherical. The similar feature is found in the

$DWR_{XKa}$–$DWR_{KaW}$ field. Particle populations with similar characteristic sizes (color circles in Figs. 1b and 1d) but different aspect ratios (0.3 vs 0.8), however, produce quite different values of DWR for both frequency pairs. It is worth also mentioning that a "soft" spheroid particle model produces $DWR_{KKa}$–$DWR_{KW}$ correspondences that are similar to those with more sophisticated models (Fig. 1a vs Fig. 1b).

In addition to power measurements, profiling cloud radar can also measure the mean Doppler velocity (MDV). Although the MDV is affected by the vertical air motion, the community has developed robust methodologies to use MDV to improve discrimination between the particle types and ice growth processes (e.g., Orr and Kropfli, 1999; Luke et al., 2010; Protat and Williams, 2011; Kalesse et al., 2013; Schrom and Kumjian, 2016; Oue et al., 2018). Particle fall speed, which

is sensitive to rime fraction, is a valuable variable to use to identify particle types (e.g., Locatelli and Hobbs, 1974; Kajikawa, 1989; Mason et al., 2019). However, using only MDV and reflectivity measurements would not be enough to distinguish between aggregation and early stage of riming, because both are associated with very similar fall speeds (e.g., Oue et al., 2016). This study, first introduces, the use of DWR coupled with MDV to identify particle types that have different degree

of riming. Figure 2 shows $DWR_{KaW}$ as a function of MDV and differential MDV (dMDV = Ka-band MDV–W-band MDV). The MDV-DWR correspondence is also sensitive to particle size distribution (PSD) details and rime degree. Fig. 2 indicates only a weak dependency on PSD, which can be advantageous for distinguishing particle types as PSD influences are minimized.

Similar to DWR and MDV, polarimetric radar observables are also sensitive to microphysical properties such as particle type, characteristic size, rime fraction, aspect ratio, canting angle, and complexity of shape (e.g., Myagkov et al., 2016). These properties provide a constraint on the particle shapes (i.e., aspect ratio) and help to mitigate the uncertainty in the DWR analysis mentioned above. The polarimetric variables are particularly suitable to identify depositional,

aggregation, and riming growths (e.g., Oue et al., 2016). The most common characteristics of the polarimetric observables representing the depositional growth are enhancements of differential reflectivity ($Z_{DR}$) and specific differential phase ($K_{DP}$) in a dendritic/plate-like growth regime (e.g., around temperature of -15°C), where the ice crystals with small aspect ratios are formed by depositional growth. The $Z_{DR}$ values decrease with decreasing height in a region of aggregation,

while ($K_{DP}$) often has a maximum just below the $Z_{DR}$ peak. With further height decrease, the aspect ratios of the individual particles increase (e.g., Vivekanandan et al., 1994; Ryzhkov et al., 1998; Kennedy and Rutledge, 2011; Andrić et al., 2013; Bechini et al., 2013; Schrom et al., 2015; Kumjian et al., 2016; Griffin et al., 2018; Matrosov et al., 2020). Similar vertical changes of the polarimetric variables have been often found in the rime-dominated regions (e.g., Zawadzki et al.,

2001; Oue et al., 2016; Giangrande et al., 2016; Kumjian and Lombardo, 2017), as heavy riming increases particle aspect ratios. Mean particle aspect ratios can be quantitatively estimated using proxies of radar circular depolarization ratios (e.g., Matrosov et al., 2017). Radar depolarization ratios can also be used to distinguish among ice hydrometeor types effectively separating oblate (e.g., plates, dendrites) from prolate (e.g., columns, needles) habits (e.g., Matrosov, 1991;



Reinking et al., 2002; Matrosov et al., 2012; Oue et al., 2015). Schrom and Kumjian (2016) suggested that a complementary use of mean vertical Doppler velocity could help to distinguish the riming process from aggregation-dominated regions. A joint analysis of polarimetric variables and Doppler spectra by Oue et al. (2018) illustrated a capability of particle type identification in Arctic mixed-phase clouds. However, distinguishing between aggregation and early stage of

riming is still challenging even though MDV and polarimetric variables are jointly used due to the similar signatures (e.g., Oue et al., 2016).

        Winter storms in the Northeast U.S often effect heavy snowfall and destruction of life and property. The lack of understanding of ice microphysical processes in the storms and poor representation of

the ice microphysics parameterizations in numerical cloud models has resulted in large uncertainty in forecasting snowfall. The ice microphysical processes including depositional, riming, and aggregation growths often coexist in the snowstorm cloudy volumes (e.g., Kumjian and Lombardo, 2017; Colle et al., 2014), making it difficult to identify the processes in the observations. To facilitate studies of cloud microphysics and dynamics in the Northeast U.S., the Stony Brook Radar

Observatory (SBRO) was established in March in 2017 in Stony Brook University, Stony Brook, NY (Fig. 3). The flagship radar of the SBRO is a very sensitive, sophisticated, and well-calibrated Ka-band (35-GHz) scanning fully-polarimetric radar (KASPR). The radar measurements are complemented by two profiling radar systems operating at W-band (94-GHz, ROGER) and K-band (24-GHz, MRRPro) and ground-based in-situ sensors. The SBRO radar systems have

collected vertically pointing triple frequency reflectivity and Doppler velocity data which were complemented by polarimetric variables from KASPR for a snowstorm observed on February 20, 2019. The triple frequency measurements showed that the DWR from the dual-wavelength measurements in conjunction with MDV and polarimetric observations had a higher efficiency to distinguish ice particle types and growth processes rather than the DWR-only diagrams from triple-

frequency measurements. This study first illustrates the capability and advantage of the use of MDV and polarimetric radar observables in conjunction with DWR measurements to identify particle types and growth processes in winter storms.




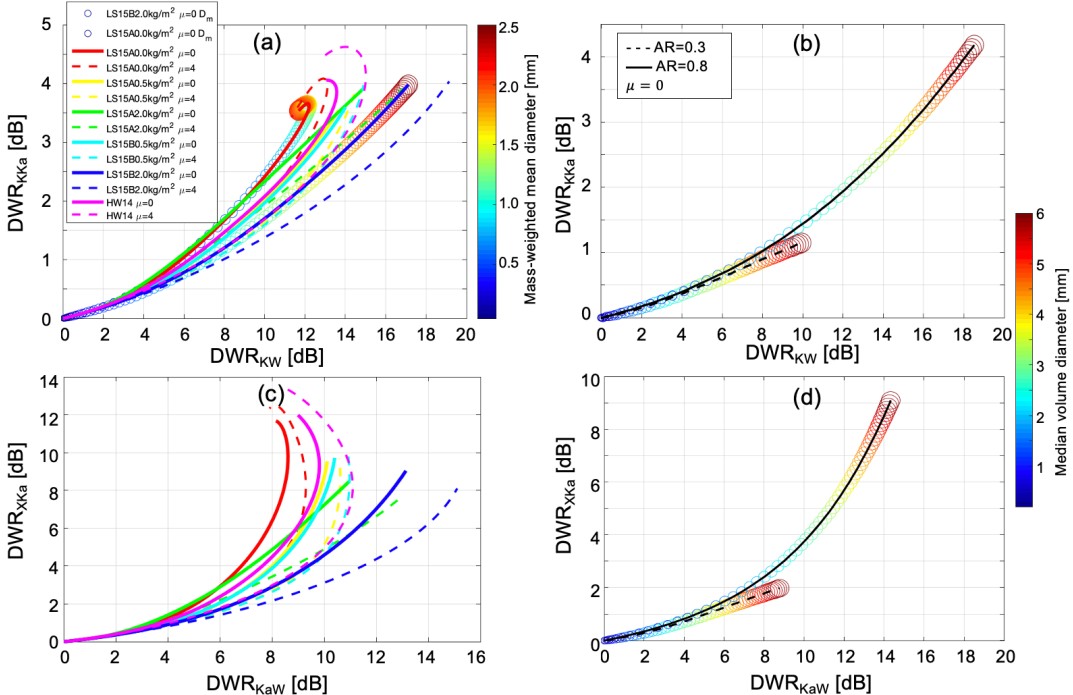

Figure 1: DWR of K-band (24 GHz) reflectivity to Ka-band (35.5 GHz) reflectivity versus that of K-band reflectivity to W-band (94 GHz) reflectivity diagram from (a) the SSRGA scattering property database and (b) Matrosov et al. (2019) accounting for particle aspect ratio (AR). (c) and (d) are the same as (a) and (b), respectively, but for DWR of X-band (10.7 GHz) reflectivity to Ka-band reflectivity versus that of Ka-band reflectivity to W-band reflectivity. Line colors in (a) and (c) represent particle models listed in Table A1. Solid and dashed lines in (a) and (c) represent the PSD's shape parameter ($\mu$) equal to 0 and 4, respectively. Color of circles in (a) represents water-equivalent mass-weighted volume diameter ($D_m$) of each PSD used to calculate DWRs; here, $D_m$ values for the particle models of Leinonen and Szyrmer's (2015) unrimed aggregates with $\mu = 0$ (solid red line) and aggregates with high rime degree (solid blue line) are presented. Solid and dashed lines in (b) and (d) represent AR=0.8 and AR=0.3, respectively. The DWRs in (b) and (d) were calculated for PSDs with $\mu = 0$ and median volume particle size (color of circles) ranging from 0.2 to 6.0 mm.

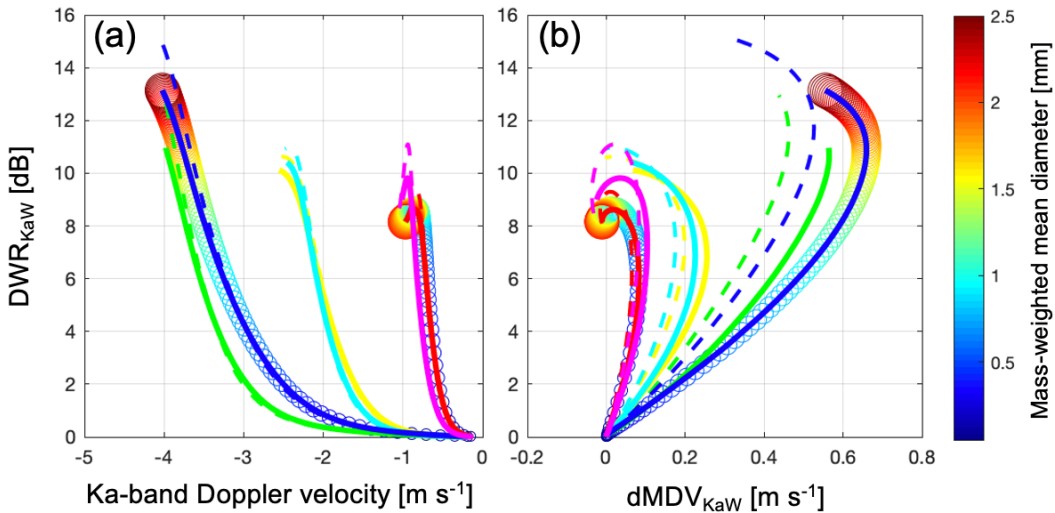

Figure 2: (a) DWR of Ka-band (35.5 GHz) reflectivity to W-band (94 GHz) reflectivity versus Ka-band mean Doppler velocity from the SSRGA scattering database with particle fall velocity models of Hogan and Westbrook (2017). (b) DWR of Ka-band reflectivity to W-band reflectivity versus difference between Ka-band MDV and W-band MDV. Negative Doppler velocity in (a) represents a downward motion. Color scale and line legends are the same as in Figure 1. These particle models can be classified into low (red and magenta), middle (yellow and cyan), and high (blue and green) rime degree particles.

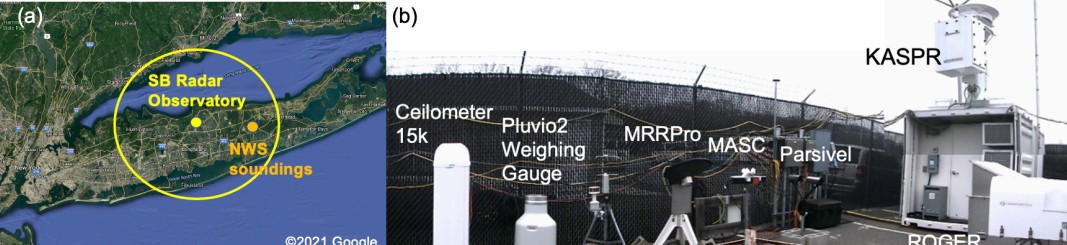

Figure 3: (a) Location of the Stony Brook Radar Observatory. (b) Instruments deployed at SBRO. A big yellow circle in (a) represents the KASPR 30-km observation range.

## 2. Data

SBRO has been in operation since March in 2017 (-73.127E, 40.897N, ~22 km west of a National Weather Service sounding site at Upton, NY; https://you.stonybrook.edu/radar/). KASPR, ROGER, and MRRPro at SBRO collected triple frequency data during a snow event on February 20, 2019. This is so far the only case where the high-quality, well-calibrated triple frequency measurements together with ground-based in-situ measurements for snow particles for the evaluation are available. The SBRO site also has ground-based in-situ observation instruments.


The in-situ instruments including a Parsivel optical disdrometer and Multi-Angle Snowflake
Camera (MASC, Garrett et al. 2014) were used to evaluate the radar-based particle identifications.
The SBRO operates ceilometers at the SBRO and Brookhaven National Laboratory sites.
Ceilometer backscatter is sensitive to cloud liquid droplets embedded in ice precipitation clouds.
A microwave radiometer was also installed at the SBRO site; however, it was not functional during
the precipitation in this study due to significant snow accumulation on the sensor.

### 2.1. Ka-band Scanning Polarimetric Radar (KASPR)

KASPR, a state-of-the-art cloud scanning radar, is capable of collecting Doppler spectra and radar
moments through alternate transmission of horizontally (H) and vertically (V) polarized waves
and simultaneous reception of co-polar and cross-polar components of the backscattered wave with
the beamwidth of 0.32°. Therefore, a full set of polarimetric radar observables is available
including radar reflectivity ($Z_{HH}$), differential reflectivity ($Z_{DR}$), differential phase ($\varphi_{DP}$), co-polar
correlation coefficient ($\rho_{hv}$), linear depolarization ratio ($L_{DR}$), and cross-polar correlation
coefficient ($\rho_{hx}$), along with Doppler velocity and spectral width. Specific differential phase ($K_{DP}$)
is estimated using an iterative algorithm proposed by Hubbert and Bringi (1995). The data post-
processing details are described in Oue et al. (2018). The KASPR was calibrated using a corner
reflector technique, providing reliable reflectivity data to calibrate the other two radar data. The
detailed configurations are also available in Kumjian et al. (2020) and Kollias et al. (2020b).

During the radar measurements on February 20, 2019, KASPR executed a scanning strategy that
consisted of surveillance (PPI) scans at a 15° elevation angle, a zenith pointing PPI, hemispheric
range-height indicator (HSRHI) scans at four azimuth angles, and a 5-minute vertically pointing
mode (VPT) during which Doppler spectrum data were collected. This pattern was repeated and
took approximately 15 minutes to complete. During a 15-min cycle, two 15°-PPI scans were
included, so that we had the 15°-PPI scans every ~7 min which were used to produce quasi-vertical
profile products. The PPI and HSRHI scans were performed in a full polarimetry mode with scan
speeds of 6 s$^{-1}$ and 2 s$^{-1}$, respectively, to collect data with a 30-m range-gate spacing, 0.6° PPI
azimuthal spacing and 0.3° HSRHI elevation spacing. The VPT mode was executed with only
horizontally polarized waves transmitted and both horizontally and vertically polarized waves
received. During the VPT mode, the Doppler spectra were collected every second with a 15-m
range-gate spacing and 0.04 m s$^{-1}$ velocity bin spacing. The zenith PPI scans were used to estimate
a systematic bias of $Z_{DR}$. The $Z_{DR}$ values presented in this study were corrected for the systematic
biases.

### 2.2. 94-GHz (W-band) Frequency Modulated Continuous Wave (FMCW) profiling radar (ROGER)

The system was initially developed as an airborne radar and was integrated on the Center for
Interdisciplinary Remotely-Piloted Aircraft Studies Twin Otter aircraft (Mead et al., 2003). In
2017, the system was refurbished with installing 24-inch parabolic dish antennas, and all the
CFMCW electronics, including a new metal frame to hold the antennas, the server computer and
the power supplies, to make it suitable for ground-based observations and easy shipping.



This radar system is capable of collecting Doppler spectra with spatiotemporal resolutions similar to KASPR (Table 1) and located next to KASPR, which allows good beam matching and reliable DWR measurements. The data during the cases were collected every four seconds at 30-m vertical spacing with a beamwidth of 0.3° (Table 1).


### 2.3. Micro Rain Radar Pro (MRRPro)

The MRRPro is the latest version of the MRR developed by Metek GmBH that has evolved to be a powerful standalone profiler for investigations of precipitation and cloud dynamics with very low installation and logistics effort. The MRRPro features a high-performance processing unit which significantly improves the options in the operating parameters (Table 1). During the observation in this study, the MRRPro collected Doppler spectra at a 60 m range-gate spacing every four seconds up to the maximum observation range of 7 km. The Nyquist velocity was 12.08 270 m s$^{-1}$ during the observations producing the velocity bin spacing of 0.192 m s$^{-1}$.

### 2.4. Ground-based in-situ measurements

A Parsivel optical disdrometer measures terminal velocity and horizontal size of individual precipitation particles passing through a sheet of light (30-mm wide, 1-mm high, and 180-mm long) with a 650-nm laser diode with a power of 3 mW (Löffler-Mang and Blahak, 2001). The total measuring surface has an area of 54 cm$^2$. The measured size and velocity are classified into one of 32 size bins ranging from 0.062 to 24.5 mm and 32 velocity bins ranging from 0.04 to20.5 280 m s$^{-1}$ every 1 minute.

The Multi-Angle Snowflake Camera (MASC) is located adjacent to the Parsivel. The MASC consists of three cameras that are separated by an angle of 36°, each pointed towards the focal point about 10 cm away (Garrett et al., 2012; 2014). On top of each camera rests a 2700 lumen 285 light emitting diode. The focal point lies within a ring that has two near-infrared-emitter-detector-pairs arranged in arrays that are separated vertically by 32 mm. The arrangement of the emitter-detector pairs allows for a trigger depth of field of 3100 mm$^2$ but because of the camera field of view and depth of focus, only about 11% of the images taken are in focus. Falling hydrometeors larger than 0.1 mm are recorded and their fall speed is calculated as the time difference between 290 triggering each emitter-detector pair.

Table 1: Specifications for KASPR, ROGER, and MRRPro.

|  | Ka-band Scanning polarimetric radar (KASPR) | W-band profiling radar (ROGER) | Micro Rain Radar Pro (MRRPro) |
|---|---|---|---|
| Frequency | 35.29 GHz (wavelength ~8.5 mm) | 94.8 GHz (wavelength ~3.2 mm) | 24.23 GHz (wavelength ~12.4 mm) |
| Range resolution | configurable between 15 – 200 m; 15 m in VPT, 30 m in RHI and PPI for this study | 5 – 150 m, 30 m for this study | > 10 m, 60 m for this study |
| Beam width | 0.32° | 0.3° | 1.5° |
| Maximum range | Configurable; 15 km in VPT, 30 km in RHI and PPI for this study | Configurable; 18.5 km for this study | Configurable; 7 km for this study |



| Velocity resolution | Configurable; 0.04 m s$^{-1}$ for this study | Configurable; 0.08 m s$^{-1}$ for this study | Configurable between 0.05 – 6.00 m s$^{-1}$ for this study |
|---|---|---|---|
| Observables | Reflectivity, Doppler velocity, full set of polarimetric variables, Doppler spectra | Reflectivity, Doppler velocity, Doppler spectra | Reflectivity, Doppler velocity, Doppler spectra |

## 3. Method

### 3.1. Reflectivity calibration and DWR estimation

KASPR reflectivity measurements were well calibrated using a corner reflector technique (Lamer et al. 2020). Therefore, systematic offsets for the MRRPro and ROGER total reflectivities have been corrected by comparing them with the KASPR reflectivity at cloud bases from a different non-precipitation cloud case. The MRRPro and ROGER reflectivity and mean Doppler velocity data were interpolated into the KASPR VPT data resolution (15-m range and 1-sec time spacings).

Gaseous attenuation needs to be considered and corrected when using short-wavelength radars (Lamer et al., 2021). The MRRPro's K-band (24 GHz) frequency is the lowest in the present study, however, the 24 GHz frequency is very close to a peak in the water vapor absorption spectrum (e.g., Liebe et al., 1993; Rosenkranz, 1998). Therefore, the water vapor attenuation for MRRPro could also be significant. We corrected the MRRPro, KASPR, and ROGER reflectivities for water vapor attenuation based on the Rosenkranz (1998) results, using sounding profiles launched twice daily (00 and 12 Z) at Upton, 21 km east of the observatory. The estimated column-integrated two-way attenuations at K, Ka, and W bands for our case study were up to 0.7, 0.2, and 1.2 dB, respectively.

Another source of the gaseous attenuation we should consider is oxygen (e.g., Liebe et al., 1993). Although the attenuation in oxygen may not be as large as that in water vapor, it may be non-negligible. We also estimated the attenuation by oxygen (i.e., dry air) for each of the three frequencies using the sounding profiles and corrected the MRRPro, KASPR, and ROGER reflectivities. The estimated column-integrated two-way attenuations for dry air at K, Ka, and W bands were generally 0.1, 0.2, and 0.3 dB, respectively.

Liquid water, which was expected to be present in precipitating clouds as supercooled droplets producing riming can also be a cause of significant attenuation. Riming commonly occurs in snowstorms observed along the U.S. North East Coast indicating the presence of significant amounts of supercooled cloud water in the snowstorms (e.g., Colle et al., 2014). However, it was difficult to identify liquid cloud layers and liquid water content and estimate specific attenuation at each range bin in the ice clouds. Moreover, attenuation by ice particles might be significant if the large amount of ice were produced in the clouds and the radar beam passed through the ice layers. Tridon et al. (2020) proposed a relative path-integrated attenuation (PIA) technique to retrieve liquid-water content using DWR profiles. A key idea of this technique is that the DWR from dual frequency radars near cloud tops, where it is expected that small ice crystals are in the Rayleigh scattering regime for both radar wavelengths, is mainly due to the PIA associated with liquid cloud droplets and ice particles. The DWR attributed to the total attenuation cshould then be equal to the DWR plateau near the cloud top. We applied the Tridon's et al. (2020) technique





to the DWR from KASPR and ROGER measurements to find DWR plateau near the cloud top
with the following adaptations:

- the measured $DWR_{KaW}$ are averaged over 450 m (30 gates) and 20 seconds (20 rays).
- The DWR variance within the moving windows defined above must be lower than 4 $dB^2$
- KASPR reflectivity and its variance (within the same moving windows) must be lower than 5 dBZ and 2.5 $dB^2$, respectively.
- the DWR plateau is found where the DWR gradient is lower than 1 dB $km^{-1}$ near the cloud top at each profile.
- the masked $DWR_{KaW}$ is averaged at the cloud-top layer, and the DWR value is considered
the total PIA.

The ROGER reflectivity was corrected for the estimated PIA linearly in the cloud layer from the ice cloud base, so that the total attenuation in the column was consistent with the estimated PIA. This assumption might produce an uncertainty, however, this kind of correction showed reasonable
results compared to no correction for PIA, as demonstrated by previous studies (e.g., Dias Neto et al., 2019; Oue et al., 2018). The DWR plateau-based PIA estimation technique requires enough sensitivity to capture cloud tops where Rayleigh scattering is expected for both. The MRRPro is sufficiently sensitive to precipitation (Fig. 4c) but not to small particles with reflectivity < 0 dBZ. The MRRPro reflectivity near its echo top could still include non-Rayleigh scatterings at Ka or W
bands. Because of this, attenuations by hydrometeors in the KASPR and MRRPro reflectivity fields were not accounted for using the DWR plateau-based attenuation correction in this study. Moreover, the presence of supercooled liquid droplets might cause total signal extinction. A microwave radiometer deployed at the SBRO observed liquid water path (LWP) values, which were generally around 150 g $m^{-2}$ before the precipitation onset. According to Tridon et al. (2020),
this amount of liquid should produce a path integrated attenuation less than 1 dB in the KASPR and MRRPro reflectivity measurements.

Another error source of the DWR analysis is radar beam mismatching. The three radars were located at the same observation site; the distance between those radars were less than 5 m,
therefore, we expect that the beam mismatch due to location is small. On the other hand, a difference in beamwidths (Table 1) is another possible cause of beam mismatching. The KASPR and ROGER beamwidths are well matched, while MRRPro's beamwidth is 5 times larger than those of the KASPR and ROGER. The beamwidth differences between MRRPro and KASPR, and MRRPro and ROGER might result in larger variabilities of DWRs.


*3.2. Mean Doppler velocity*

Similar to the reflectivity measurements, the MRRPro and ROGER mean Doppler velocity data
were interpolated into the KASPR VPT data resolution. The observed mean Doppler velocities from the 3 radars were corrected for air density changes based on the sounding profiles and adjusted to the surface.

*3.3. KASPR polarimetric observables*



The polarimetric radar observables such as $Z_{DR}$ and $K_{DP}$ are more prominent when they are collected at lower elevation scans, whereas the DWR data were collected by vertically pointing measurements. To compare those two data sets from the different types of scans, we employed a quasi-vertical profile (QVP) technique proposed by Ryzhkov et al. (2016). The QVP technique azimuthally averages polarimetric radar variables for each conical PPI scan at non-zero elevations to produce these variables in a height versus time format. The QVPs have high vertical resolutions allowing for capture important polarimetric radar signatures and their evolution (e.g., Griffin et al., 2018; 2020; Kumjian and Lombardo, 2017; Troemel et al., 2019). We use the PPI scans at an elevation angle of 15° every 7-8 min with a scan rate of 6° s$^{-1}$. Since the slant range resolution of the 15° PPI data is 30 m, the corresponding QVP data have the vertical spacing of approximately 10 m and the maximum height of 7.8 km. Note that the actual vertical resolution of QVP is determined by the vertical size of the radar resolution volume, which increases with distance from the radar (Ryzhkov et al., 2016). The use of conical PPI at a higher elevation angle (15°) for QVP reconstruction ensures relatively high horizontal resolution at lower altitudes (11 km at the height of 2 km) that facilitates direct comparison with the DWR profiles from the three radar vertically pointing measurements. The KASPR QVP data were interpolated into the KASPR VPT data resolution, similar to Oue et al. (2018). Because a single PPI scan was performed every 7 min while the KASPR 5-min VPT dwell collecting profile data every second was performed in a 15-min interval, a single KASPR QVP corresponds to about 150 DWR profiles.

## 4. Case Description

During the 2018-2019 and 2019-2020 winter seasons, most of precipitation was non-dry snow including rain, wet snow, refrozen particles, and sleet, and provided very few dry snow events at the ground around Long Island, NY. Those non-dry snow particles caused significant attenuation of radar signals particularly at millimeter wavelengths and accumulation on the radomes. Although the majority of the observed precipitation cases during the winter seasons included the non-dry snow particles near the ground, for a few cases before snow started to accumulate, ice clouds (with, possibly, embedded supercooled cloud layers) were observed aloft. We selected a period from a snow precipitation case on February 20, 2019, when KASPR VPT, ROGER, and MRRPro simultaneously observed snowfall without significant attenuations.

A high pressure system at the surface persisted around Long Island from 09 UTC to 21 UTC on February 20, 2019, while two troughs were also identified to the southeast of Long Island: one was elongated from a low pressure system in Tennessee to Pennsylvania, and the other was associated with another low pressure system around the coast of Georgia and lay along the East Coast toward Long Island. Either one of the two or both could be accompanied by a warm frontal-like stratiform precipitation providing snow in Long Island. Snow precipitation started at around 18:00 UTC at SBRO. Based on the MASC-observed particle images and Parsivel-observed particle diameter and fall velocity, dry snow aggregates dominated from the beginning till 23:30 UTC, and then the dominant precipitation included mixed-phase particles and changed into pure rain at around 04:00 UTC on February 21.



Figures 4 and 5 show the time-height curtain images of the reflectivities from the three radars and KASPR and ROGER MDV together with KASPR polarimetric QVPs. The triple frequency measurements started at 15:41 UTC. The cloud base descended until the lidar backscatter signal reached the KASPR's lowest gate (0.4 km altitude) at 19:00 UTC. The cloud top attained 10 km
altitude, but the cloud top was decoupled from the ice precipitation since 17:45 UTC.

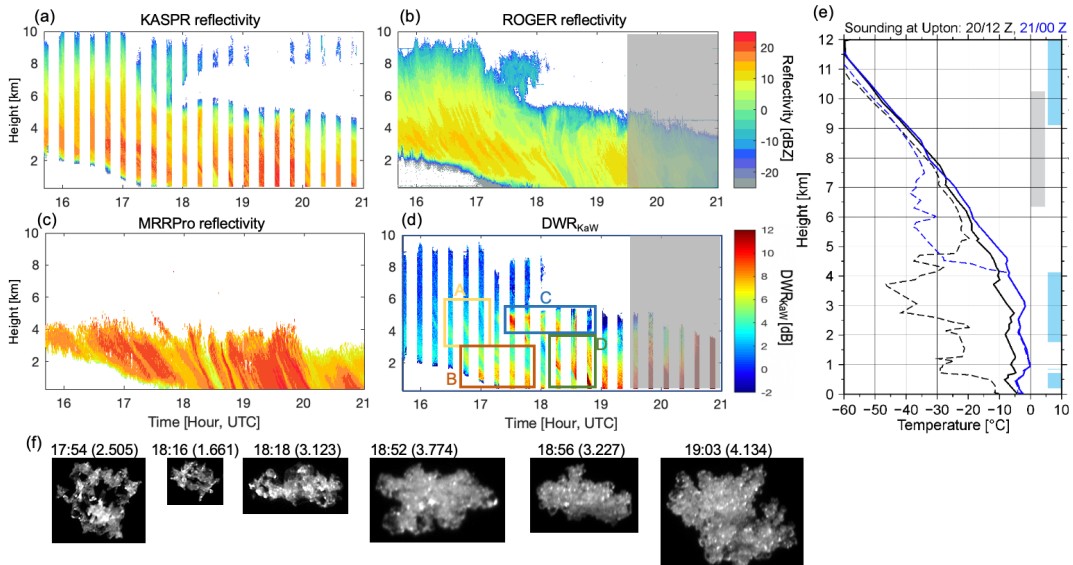

Figure 4: Height-time cross sections of (a) KASPR VPT reflectivity, (c) ROGER reflectivity, (b)
MRRPro reflectivity, and (d) DWR of KASPR reflectivity to ROGER reflectivity on Feb. 20, 2020, (e) vertical profiles of temperature (solid line) and dew point temperature (dashed line) from the NWS Upton sounding measurements at 12 UTC on Feb. 20 (black color) and 00 UTC on Feb. 21 (blue color), 2020, and (f) examples of snowflake images captured by MASC. Boxes in (d) represent analysis regions used for Figs. 7-10. Gray and blue shades in (e) represent
regions of supersaturation with respect to ice for 12 UTC on Feb. 20 and 00 UTC on Feb. 21, respectively. Each image in (f) displays observation time and maximum dimension in parenthesis (unit is mm).





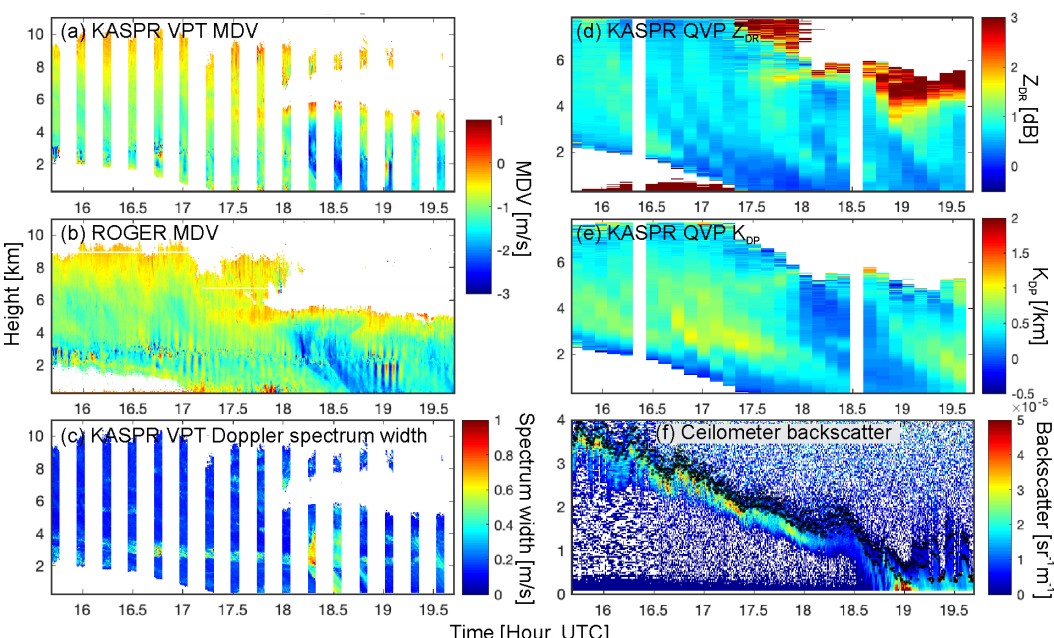

Figure 5: Height-time cross sections of (a) MDV from the KASPR VPT measurements, (b) MDV from the ROGER measurements, (c) spectrum width from the KASPR VPT measurements, (d) QVP of KASPR $Z_{DR}$, (e) QVP of KASPR $K_{DP}$, and (f) ceilometer backscatter on Feb. 20, 2020. Black dots in (f) represent cloud base heights.

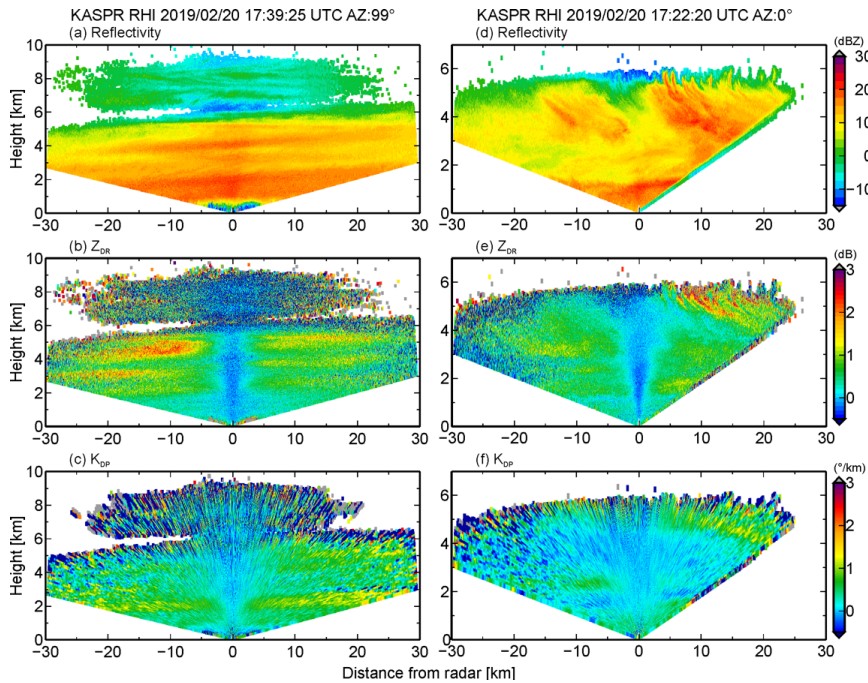



Figure 6: KASPR (a,d) reflectivity, (b,e) $Z_{DR}$, and (c,f) $K_{DP}$ from RHI measurements (a,b,c) at 17:39 UTC at an azimuth angle of 99° and (d,e,f) at 18:22 UTC at an azimuth angle of 0°.

The KASPR and ROGER reflectivity fields indicated generating cell-like features by 17:10 UTC near the cloud top above 8 km altitude (Fig. 4), as the MDV indicated convection features in the generating cells (Fig. 5). These generating cells produced fallstreaks underneath as reflectivity increased toward the ground and reached the ground by 18:20 UTC. KASPR RHI scans in Fig. 6 showed fallstreaks elongating from the generating cell layer following the wind direction above

the 2 km altitude (W-E direction). The KASPR $Z_{DR}$ was enhanced between the fallstreaks while $K_{DP}$ increased in the lower part of the enhanced $Z_{DR}$ layer and just below the enhanced $Z_{DR}$ layer. The enhancement of $Z_{DR}$ generated at a height of 5-6 km, where the temperature was ranging from -20°C to -15°C corresponding to a dendritic growth layer and close to supersaturation with respect to ice from the 12 UTC sounding (Fig. 4e). This is a typical signature of the aggregation and

generation of dendritic crystals commonly observed by previous studies (e.g., Kennedy and Rutledge, 2011; Schneebeli et al., 2013; Kumjian et al., 2014; Williams et al., 2015; Oue et al. 2018). The $DWR_{KaW}$ increased toward the ground in the fallstreaks, as reflectivity increased. At times corresponding to the fallstreaks reaching the ground, MASC observed large aggregates.

Starting from 17:50 UTC, precipitation observed at the surface originated at 6 km altitude. The KASPR RHI measurements revealed that cloud aloft was decoupled from below and there were structured generating cells near the lower cloud top at 6 km (Fig. 6). Large $Z_{DR}$ values were observed between the generating cells and between fallstreaks, while $K_{DP}$ slightly increased just below the generating cell layer but decreased to near zero within the fallstreaks. There was a layer

of large $DWR_{KaW}$ at 4-5.5 km altitude from 17:15 to 18:50 UTC, even though the Ka-band reflectivity was smaller than that in the former fallstreaks. The large $DWR_{KaW}$ extended toward the ground and reached the surface at 18:30 UTC (Fig. 4d), as the KASPR polarimetric signatures associated with the fallstreaks reached the surface (Figs. 5d and 5e). Corresponding to the time when the fallstreak features reached the surface, MASC observed rimed particles (Fig. 4f). These

DWR and polarimetric features likely indicate an ice particle growth, however, it is hard to determine specific ice growth processes (i.e., distinguishing riming and aggregation processes) from the $DWR_{KaW}$ or the polarimetric observables only.

    There are several signatures that suggest different types of ice particle growth during the two

periods. A distinct difference between the two periods is found in the MDV from the vertically-pointing measurements and the KASPR polarimetric observables; they suggest different ice particle fall speeds attributed to the particle types and microphysics. The downward motion within the fallstreaks during the first period gradually increased toward the ground to 1.5 m s$^{-1}$, indicative of growth of individual ice particles. The fallstreaks corresponded to the enhanced $K_{DP}$, but

decreased $Z_{DR}$, suggesting that oblate small particles aggregated within the fall streaks. In contrast, the latter period corresponded to decreased $K_{DP}$, while $Z_{DR}$ values are enhanced near the 6 km altitude but decreased toward the surface. These $K_{DP}$ and $Z_{DR}$ evolutions suggest that small oblate ice crystals formed at 6 km altitude and aggregated as they fell forming spherical shapes, as many previous polarimetric radar studies observed. The MDV showed faster downward motion

compared to the fallstreaks in the first period, suggesting heavy aggregation and/or riming.



Another interesting characteristic to be noted is that there was a distinct region of turbulence, which can clearly be seen as a layer with large spectrum width and variability of MDV at around 3 km altitude. This was consistent with large lidar backscatter values suggesting liquid cloud base. The reflectivity and DWR of fall streaks were intensified below the turbulence layer.


Although the individual radar parameters suggest a variety of ice particle types and microphysical processes, it is not straightforward to identify the ice particle types and distinguish the processes, in particular aggregation and riming, by a single measurement.


## 5. Results and Discussions

### 5.1. DWRs from the three frequencies


Based on the $DWR_{KaW}$ height-time plots, we selected four regions as shown in Fig. 4d, each of which had similar DWRs, MDV, and polarimetry features to identify ice particle types and their growth processes. We first present traditional triple-frequency DWR-DWR diagrams ($DWR_{KW}$ versus $DWR_{KKa}$ in Fig. 7) for each selected region. The DWRs from Region A and Region B tend to be distributed toward the model low rime degree lines (larger $DWR_{KKa}$ at a given $DWR_{KW}$), while those from Region C and Region D were distributed toward the higher rime degree regions (smaller $DWR_{KKa}$ at a given $DWR_{KW}$). These are consistent with MASC ice particle observations. Although the distribution of the DWRs for each region seems to be significantly separated, most of the data are overlapped, making it hard to distinguish the growth processes and types. This is, in part, because K-band (24 GHz) and Ka-band (35 GHz) measurements are not sufficiently separated in the frequency domain.



Besides the insufficient frequency separation, there are data points that deviate from the model lines in the $DWR_{KW}$ versus $DWR_{KKa}$ field in each region. There are several causes of such deviations (e.g., Lamer et al., 2020). The most likely cause is unaccounted attenuations particularly at Ka and W bands due to either supercooled cloud water or ice or both. The ceilometer backscatter measurements shown in Fig. 5f, in addition to the MWR LWP measurements, suggest that thin supercooled liquid cloud layers were indeed present at least around the large Doppler spectrum width layer. Unfortunately, the ceilometer backscatter information is insufficient to provide a complete mapping of such layers because of complete signal extinction caused either by the ice clouds or by underlying liquid layers themselves. Ice particles could also cause signal attenuation (Battaglia et al., 2020; Tridon et al., 2020) particularly for the shorter-wavelength radars. Although the DWR plateau-based PIA technique has corrected the ROGER reflectivity for those attenuations related to the KASPR reflectivity (Sect. 3.1), the attenuation in the KASPR reflectivity itself cannot be accounted for in this study. This factor also causes underestimation of the PIA-corrected ROGER reflectivity.




Secondly, the beam mismatch could be significant when the radar beams penetrate narrow, fine fallstreaks, even though the radars were collocated within 5 meters from each other. As mentioned previously, the KASPR and ROGER beamwidths are well matched (0.3°), while the MRRPro's beamwidth is 5-times larger (1.5°). The radar sampling volumes, which are larger at higher altitudes, cannot resolve the small time and spatial scale phenomena, and the difference in






beamwidth is a source of uncertainty. Moreover, the ice particle models may not represent all the gamut of ice particles possibly present in the clouds.


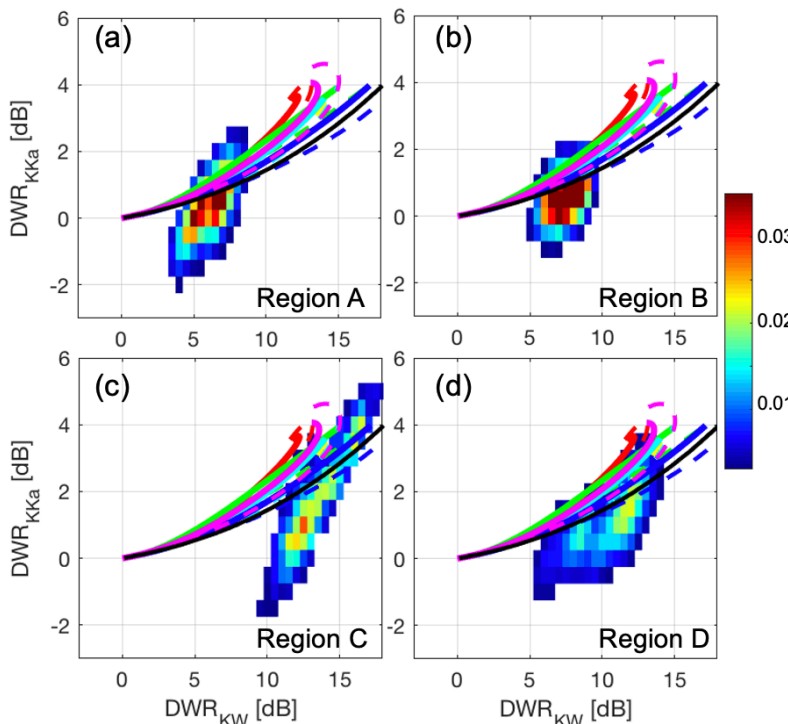

Figure 7: (a-d) DWR$_{KKa}$ versus DWR$_{KW}$ diagrams for Regions A, B, C, and D, respectively. Color shades represent normalized frequency. Lines in each panel represent the SSRGA calculations using particle type and PSD models described in Sect. 3.4. The color line legend is the same as in Figure 1a, and black lines are same as black lines in Fig. 1b.


*5.2. DWRs coupled with MDV and polarimetric variables*

Observed MDV is attributed to mainly the vertical air motion and the particle fall speeds, which is sensitive to particle size, rime degree, size distribution, and type, and can provide additional information to distinguish ice types and processes. Kneifel and Moisseev (2020) demonstrated that MDV is a function of rime fraction. We further illustrate that MDV coupled with DWR shows a good indicator of degree of riming. Figure 8 shows the observed DWR$_{KaW}$ as a function of KASPR

MDV together with the model plot with different rime degrees (lines). Most of DWR$_{KaW}$ values from Region A are less than 7 dB and are located between the middle (yellow and cyan) and low (red and magenta) rime degree lines, suggesting light riming of small aggregates (Fig. 8a). These particles could grow keeping the similar degree of riming by aggregation, as the data points from Region B are shifted toward larger DWR$_{KaW}$ values between the middle and low rime degree lines

(Fig. 8b). It is possible that the turbulence layer at around 3 km altitude (Figs. 5b and 5c) contributed to light riming. The turbulence also contributed to the wide distribution of MDV.



DWR$_{KaW}$ from Region C generally follows the low-rime degree particle lines; DWR$_{KaW}$ increased from near zero to 10 dB while MDV changed from near zero to -0.8 m s$^{-1}$. Some data points are shifted toward the middle-rime degree particle lines (i.e. faster downward motion at a given DWR$_{KaW}$). These data clusters suggest that the aggregation dominated, but some particles started riming. Region D, which is located below Region C, has also generally two data clusters. A smaller data cluster closely follows the low-rime degree lines, as the DWR$_{KaW}$ increased 2 to 9 dB while the MDV changed from -0.6 m s$^{-1}$ to -1.3 m s$^{-1}$. The other population, which has higher occurrence, is generally along the middle rime degree lines; the DWR$_{KaW}$ increased from 3 to 12 dB while the MDV changed from -1.8 m s$^{-1}$ to -2.5 m s$^{-1}$ in the middle of the population. The left edge of the second data population is closer to the higher riming degree (blue and green) lines. Those downward MDVs belonging to the two populations are consistent with fall velocities of aggregates and heavily-rimed particles, respectively, studied by Locatelli and Hobbs (1974). These characteristics suggest that aggregates produced near the cloud top at 6 km rimed during the falling as particle fall speeds quickly increased. These distinct separations of the particle populations associated with the particle growth processes are not clearly found in the triple-frequency DWR field in Fig. 7, whereas these are also shown in the DWR$_{KW}$ versus KASPR MDV (Fig. 8e) and DWR$_{KKa}$ versus KASPR MDV (Fig. 8f) diagrams.

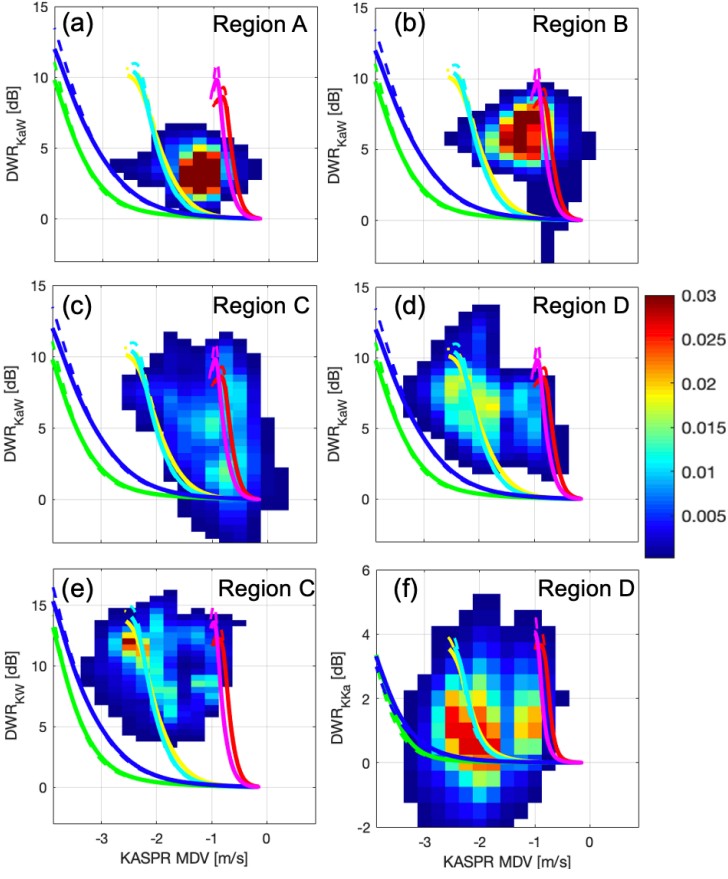


Figure 8: (a-d) KASPR VPT MDV versus $DWR_{KaW}$ diagrams for Regions A, B, C, and D, respectively. (e) KASPR VPT MDV versus $DWR_{KW}$ diagram for Regions D, and (f) KASPR VPT MDV versus $DWR_{KKa}$ diagram for Regions D. Color shades represent normalized frequency. Lines in each panel represent the SSRGA calculations using different particle models and PSDs described in Sect. 3.4. The legend of the lines is same as Figure 1.

Adding polarimetric information supports this interpretation and gives further insights into particle microphysics in terms of their shapes. Figures 9a-d and Figures 9e-h are the same as Figs. 8a-d, respectively, but the color shade represents KASPR QVP $Z_{DR}$ (Figs. 9a-d) and $K_{DP}$ (Figs. 9e-h), respectively. In the data grouping following the low rime degree lines for Region C and Region D, $Z_{DR}$ values decreased as DWR increased (Figs. 9c,d), while $K_{DP}$ values slightly decreased by approximately $0.2 °$ $km^{-1}$(Figs. 9g,h). It can be interpreted that small ice particles producing near zero DWRs were horizontally-oriented oblate particles in the dendritic crystal growth zone (temperature of -15 to -10°C), which produced large $Z_{DR}$ values and then aggregated to be large size as DWR increased. On the other hand, individual frequency pair DWR for vertically pointing measurements also strongly depends on particle aspect ratios (Matrosov et al. 2019). The impacts of particle aspect ratio on $DWR_{KaW}$ values could be as high as ~3 and ~5 dB for particle distributions with median volume size of 1 and 2 mm, respectively (Matrosov 2021). The increase of $DWR_{KaW}$ in the diagrams can include both the particle size and shape effects.

The DWR-MDV diagrams suggest that as $DWR_{KaW}$ increased, the MDV corresponding to the low rime degree particle populations in both Region C and Region D reached ~-1 m $s^{-1}$, consistent with the fall speeds of low rime degree aggregates. This effect is more likely due to the increase of size rather than aspect ratio. During the aggregation process, the size distribution of snowflakes evolves in such a way that the concentration of smaller, higher-density particles decreases whereas the number of larger, lower density snowflakes increases. This is a primary reason for the reduction of both $Z_{DR}$ and $K_{DP}$ due to aggregation, although the increase of the average aspect ratio and possibly more chaotic orientation additionally contribute to such a reduction. The $K_{DP}$ values could also be accounted for by changes in the number concentration of the horizontally-oriented oblate particles (with aspect ratio < 1); its increase contributes to increasing $K_{DP}$.

In the DWR-MDV data clusters following the high-rime degree particle lines in Region C, $Z_{DR}$ and $K_{DP}$ quickly decreased as the $DWR_{KaW}$ and the magnitude of MDV increased; $Z_{DR}$ values decreased from 2 dB to 0.5 dB, and the $K_{DP}$ values decreased from $0.4 °$ $km^{-1}$ to near zero (Figs. 9c,g). Although the increase of the $DWR_{KaW}$ includes the effects of both size and aspect ratio as discussed above, the increase of the magnitude of MDV can represent the increase of size. The $Z_{DR}$ and $K_{DP}$ values are lower than those from the cluster along the low-rime degree models at a given $DWR_{KaW}$. Lower $K_{DP}$ and $Z_{DR}$ values suggest a particle growth by heavier riming, which tend to produce more spherical particles.

These $Z_{DR}$ and $K_{DP}$ characteristics shown in both low and high rime degree particle data groups in Region C are very similar to those in Region D, but the $K_{DP}$ and $Z_{DR}$ values in Region D are generally lower at a given $DWR_{KaW}$, with a mean MDV of -3.5 m $s^{-1}$ (Figs. 9d,h). The lower $K_{DP}$ and $Z_{DR}$ in Region D represent further particle growth which is accompanied by the decrease of their density, increase of their aspect ratios, and more random orientation.



It is interesting that for DWR$_{KaW}$ values less than 5 dB in Region C, the observed Z$_{DR}$ values with faster fall speeds (corresponding to the high-rime degree particle lines) are larger than those with slower fall speeds (corresponding to the low- rime degree particle lines) at a given DWR$_{KaW}$ (Fig. 9c). This suggests that riming first worked to fill the gaps of branches of dendrite crystals, resulting in increasing the mass of individual crystals without significant change in their aspect ratio and thus Z$_{DR}$ increased. This type of riming would not significantly contribute to the increase of K$_{DP}$ (Fig. 9g), likely due to low concentration of such particles. This characteristic is consistent with the early stage of riming reported by previous studies (e.g., Oue et al., 2016; Li et al., 2018).

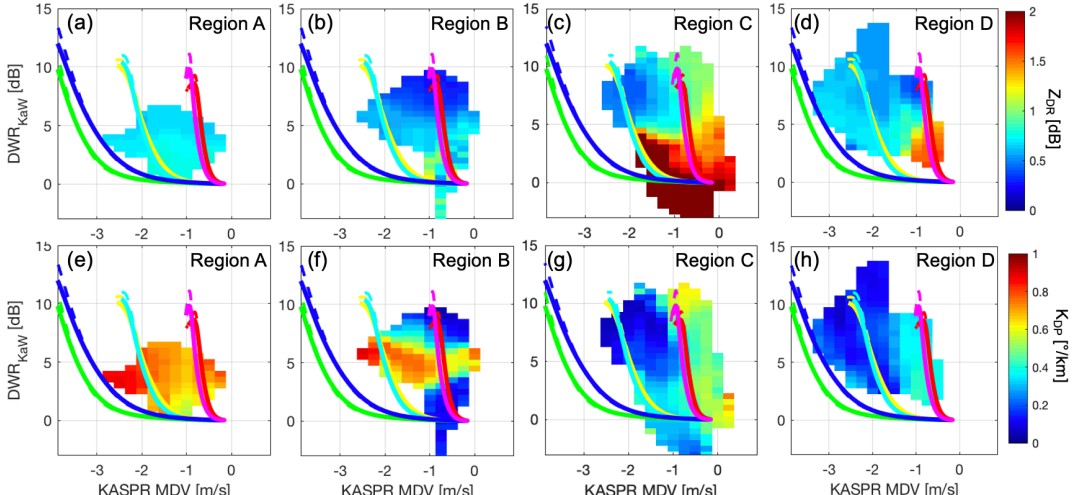

Figure 9: KASPR VPT MDV versus DWR$_{KaW}$ diagrams for (a,e) Region A and (b,f) Region B, (c,g) Region C, and (d,h) Region D. Color shades in (a-d) and (e-h) represent KASPR QVP Z$_{DR}$ and K$_{DP}$, respectively, averaged at each MDV-DWR bin. Lines in each panel represent the SSRGA calculations using particle type and PSD models described in Sect. 3.4. The legend of the lines is the same as in Figure 1.

Compared to Region C and Region D, the polarimetric observables in Region A and Region B (Figs. 9 a-b and Figs. 9e-f) do not show clear trends with changes of rime degree, and the dynamical oscillation shown in Figs. 5a-c results in an uncertainty in the particle identification for Region A, particularly when DWR$_{KaW}$ values are smaller than 5 dB and MDV is varying between -3 and 0 m s$^{-1}$. Adding polarimetric variables together with temperature information facilitates the interpretation of the microphysics. Z$_{DR}$ values in Region A are positive, but smaller than 1 dB, and smaller than those from the later fallstreaks (Region C and Region D, for a given DWR), suggesting an aggregation process which was accompanied by a decrease of particle density, an increase of their aspect ratios, and more random particle orientations compared to Region C and Region D. In contrast, K$_{DP}$ is larger than that from the later fallstreaks. The large K$_{DP}$ and smaller Z$_{DR}$ values in Region A suggest aggregation intensified by higher number concentration of ice crystals. The increase of ice number concentration can be explained by two processes. One possible cause is that near the dendrite growth regime (around -15°C), dendritic ice crystals were nucleated. The dendritic branches could work to facilitate interlocking (Pruppacher and Klett, 2010). This is


a well-known characteristic in winterstorms reported by many previous studies using polarimetric radar measurements (e.g., Kennedy and Rutledge, 2011). Another process is seeding from above (e.g., Griffin et al, 2018; Oue et al. 2018), which is more likely to contribute to the increase of ice concentration for this case. The cloud top height during observations in Region A and Region B reached 10 km, approximately 4-km higher than in Region C and Region D (Figs. 5 and 6). This fact suggests that higher concentration of ice particles aloft seeded in Region A. Moreover, a possible light riming in the turbulence region could increase the mass of individual particles, hence $K_{DP}$, as the cluster extended to the middle rime degree included large $K_{DP}$ values.

The particles were further growing at lower altitudes as $DWR_{KaW}$ increased with decreasing $Z_{DR}$ in Region B. However, a sublimation process near the ground could also be plausible. The nearest soundings at Upton (12 Z, black lines in Fig. 4e) showed a dry air condition at the lower altitudes. This sounding time was ~5-6 hours before the radar observation, but the dry air condition could still be present near the ground, thus favouring sublimation in the lower altitudes of Region B. Due to sublimation some branches and/or edges of aggregate particles could have disappeared, resulting in decreasing the mean volume diameter. The classical aggregation process could have stopped with $K_{DP}$ remaining relatively large because it usually decreases proportionally to the mean volume diameter. Decrease of IWC attributed to the sublimation might have been minor with  any noticeable impact on $K_{DP}$. These processes related to the sublimation are represented by a cluster with high $K_{DP}$ in Region B, where the $DWR_{KaW}$ values slightly increased while $K_{DP}$ values kept high compared to Region A. The sublimation also contributed to decreasing particle fall speed, as shown by a minor decrease of the magnitude of MDV in the data group, but the MDVprobably resulted from some balance between the fall speed increase due to aggregation and its decrease due to sublimation. The classical diabatic sublimation cools and moistens the ambient air. Therefore, the sublimation subsided as the cloud base descended with time (Fig. 5f) and snow particles in the fallstreaks eventually reached the ground.

### 5.3. DWR coupled with differential MDV

The MDV measurements also have frequency dependencies because of the complex interplay between non-Rayleigh effects and the PSDs. . Figure 10 shows dependencies of the ice particle types on the $DWR_{KaW}$ versus differential MDV (dMDV = KASPR MDV – ROGER MDV) diagrams for Region C and Region D, together with the scattering calculations using the particle models. Similar to the DWR-MDV diagram in Fig. 9, Region C includes a cluster with a higher number of occurrences along the low-to-middle rime degree particle lines, and a lower frequency cluster extends toward the high-rime degree particle lines. Region D has more data points for the high-rime degree particle population.

Region D also includes large dMDV values greater than 0.6 m s$^{-1}$ for $DWR_{KaW}$ values between 3 and 10 dB (Fig. 10b). It is possible that the larger values of dMDV correspond to an increase in the particle sizes and not to changes in the degree of riming. The $Z_{DR}$ values corresponding to these large dMDV values (Fig. 10c) are approximately 0.7 dB, suggesting that the particles were non-spherical, possibly contributing to the decrease of $DWR_{KaW}$ compared to the spherical particles.

As the scattering calculations show, distinguishing among different degrees of riming requires
accurate measurements of MDV with an error of few hundredths of 1 m s⁻¹ and exact range-time-
bin gate matching for lower DWR$_{KaW}$ (< 5 dB). Although the vertical air motion contributions in
MDV from each radar are cancelled out in dMDV, subgrid scale turbulence, the wide range of
particle fall speeds, and different sampling times for the observations (1 s for KASPR VPT and 4
s for ROGER) can all be sources of uncertainties. This limitation may affect the scatterplot
distributions, e.g. with some points clustering ouside the envelop of the model's lines. This
limitation also affects Region A (not shown).

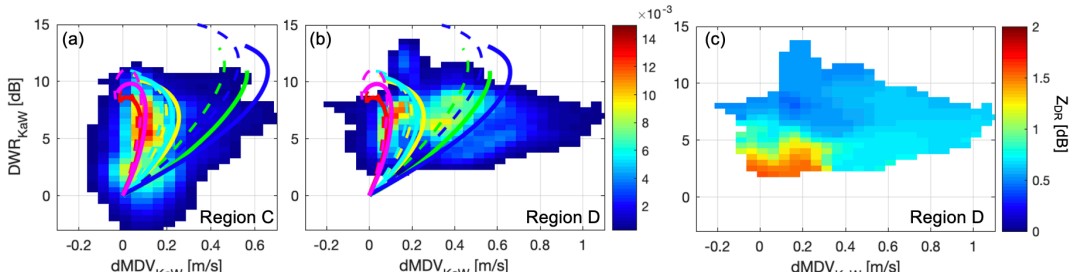

Figure 10: Difference of KASPR VPT MDV and ROGER MDV versus DWR$_{KaW}$ diagrams for (a)
Region C and (b) Region D. Color shades represent normalized frequency. (c) is the same as (b),
but the color shade represents KASPR QVP Z$_{DR}$. Lines in (a) and (b) panels represent the SSRGA
calculations using particle type and PSD models described in Sect. 3.4. The legend of the lines is
same as Figure 1.


*5.4. Evaluation using ground-based in-situ measurements*

The particle properties retrieved from the ground-based measurements including fall speed, size,
aspect ratio, and area ratio are the result of ice growth processes in the clouds aloft. The Parsivel
and MASC observations allowed us to evaluate the radar-based particle characteristics described
above. The Parsivel and MASC collocated with the radars collected precipitation particles after
18:13 UTC and 18:16 UTC, respectively. The snow images from MASC were quantified by
measurements of aspect ratio and area ratio, and their time series were presented in Fig. 11. Figures
11a and 11b present frequencies (color shade) together with median values (black line) observed
for a 20-min time range every 1 minute.

We also estimated mass-weighted mean diameter for Parsivel-measured PSD. The ice particle
mass was estimated using a methodology proposed by von Lerber et al. (2017). The methodology
is based on a theory that individual particle mass can be expressed based on a hydrodynamic theory
derived by Böhm (1989) using the Reynold's number and the Best number (e.g., Mitchell, 1996;
Mitchell and Heymsfield, 2005; Heymsfield and Westbrook, 2010). The equation of mass (Eq. 5
of von Lerber et al., 2017) indicates that the mass can be a function of fall velocity, area ratio, and
size. In the present study, the area ratio is derived from the MASC images, and the fall velocity
and size are estimated from the Parsivel measurements. The Parsivel-observed particle diameter
and fall speed are fitted to a form of $V = aD^b$ where $a$ and $b$ are constants using the 20-min





integrated data. Previous studies pointed out that Parsivel's velocity and even size measurements for snow include large uncertainties owing to the sampling limitation (Battaglia et al. 2010). Before estimating the relationships, we removed unrealistic velocity values as follows: 1) data having a small diameter (<1 mm) and too large velocity (>1.5 m s$^{-1}$) according to Locatelli and Hobbs (1974), and 2) data outside upper and lower boundaries of the V-D relationships. The upper boundary was determined based on Locatelli and Hobbs's (1974) V-D relationships for rimed aggregates, and the lower boundary was determined based on Szyrmer and Zawadzki's (2010) V-D relationships for unrimed aggregates. The Parsivel-measured size was adjusted to the maximum dimension using a technique proposed by von Lerber et al. (2017). Figure 11c presents the time series of the estimated water-equivalent mass-weighted mean diameter from the Parsivel-measured PSD.

These time series are consistent with the fallstreaks reaching the ground. Aspect ratio represents oblateness of particles relating to $Z_{DR}$ and partly contributing to $K_{DP}$. Korolev and Isaac (2003) and Jian et al. (2017) suggested that mean aspect ratio of observed snowflakes can be around 0.6 or a bit smaller, while that of heavy-rimed particles such as graupel increases toward 1. Radar depolarization-based retrievals of snowflake aspect ratios near the ground indicated mean aspect ratios of about 0.4 - 0.5 (e.g., Matrosov et al., 2020). Area ratio in the current study is defined as the ratio of the area of the snowflake, which is found by counting all white pixels in a black and white image, to the area of the circumscribing circle defined by the maximum diameter from MASC. The area ratio increases with riming (von Lerber et al. 2017).

The aspect ratio was relatively low before 18:30 UTC, where the median value was less than 0.6. At the same time, the area ratio was also relatively low, where the median area ratio was smaller than 0.5. This period corresponds to time where fall streaks included in Region A and Region B reached near the ground, consistent with the radar MDV-DWR characteristics. The mass-weighted mean size was approximately 0.4 mm, consistent with the scattering model calculations shown in Fig. 2a. It should be noted also that aspect ratio estimates from in situ data (e.g., from Parsivel and/or MASC measurements) are inferred from 2D particles projections, so these estimates usually overestimate actual aspect ratios, which are defined as true minor-to-major particle dimension ratios (Jiang et al., 2017; Matrosov et al., 2017).

The median in situ aspect ratio exceeded 0.6 between 18:30 and 19:00 UTC, while area ratio also increased. The water-equivalent mass-weighted mean diameter increased after 18:33 UTC, as it exceeded 1.3 mm between 18:38 and 19:48 UTC except 18:58, 19:22, and 19:28) UTC. Those large diameter periods correspond to times where fall streaks included in Region D reached the ground. The ground-based characteristics suggests that snowflakes were larger aggregates, which heavily rimed having faster fall speeds, consistent with the observed characteristics of the radar MDV-DWR coupled with the polarimetric observable.

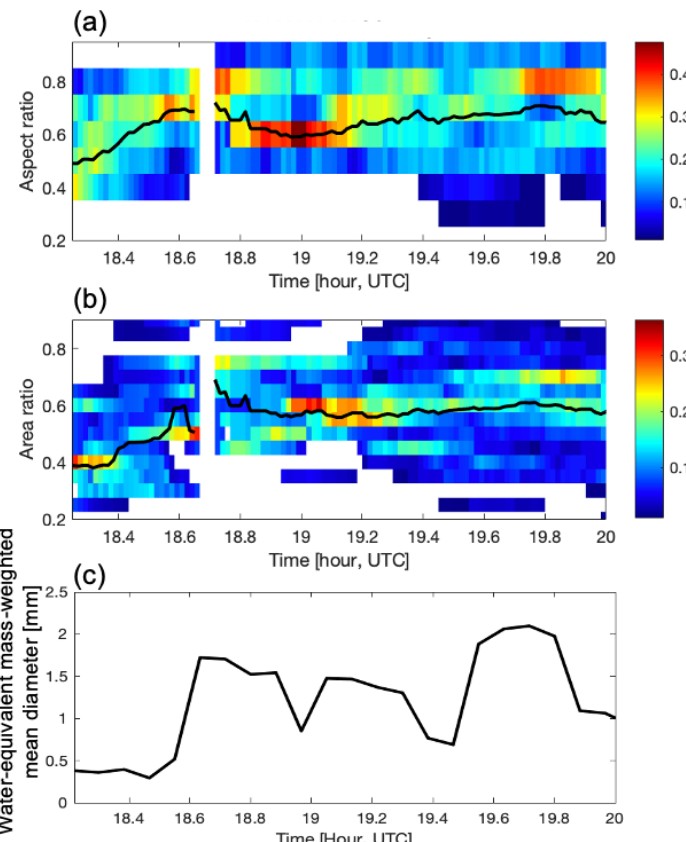

Figure 11: Time series of (a) aspect ratio and (b) area ratio of snow particles measured by the MASC and (c) water-equivalent mass-weighted mean size of Parsivel-measured PSDs. Color shades and black lines in (a,b) represent normalized frequency and median values, respectively, for snow particles collected during a 15-min window every 1 minute.

795

## 6. Summary

DWRs from triple-frequency measurements are useful to identify ice particle types and processes as proposed in previous studies . For the technique to be effective  the radar frequencies need to be well separated. This requirement limits applications when using 24, 35, and 94 GHz frequency radars, like in this study. Despite this limitation, MDV and polarimetric variables can be used complementarily to identify ice particle types and distinguish among different ice growth processes and even reveal additional microphysical details.

805



We conducted triple frequency measurements using the MRRPro (24 GHz), the Ka-band scanning polarimetric radar (KASPR, 35 GHz), and the W-band profiling radar (ROGER, 94 GHz) at the Stony Brook University Radar Observatory in the winter season of 2019-2020. We successfully collected the triple frequency data from vertically-pointing measurements for a snowstorm along the U.S. North East coast on February 20, 2019. Quasi-vertical profile (QVP) height-versus-time data were also obtained from KASPR PPI scans at an elevation angle of 15°. We investigated all pairs of DWR from the triple frequencies (i.e., $DWR_{KKa}$, $DWR_{KW}$, and $DWR_{KaW}$) in conjunction with MDV from the KASPR vertically-pointing measurements and $Z_{DR}$ and $K_{DP}$ from the KASPR QVPs. Overall, it was challenging to discern the precipitation particle types in the $DWR_{KKa}$-versus-$DWR_{KW}$ diagram only, likely due to insufficient separation of the K-band frequency from Ka band, whereas the DWR-versus-MDV diagrams for all DWR pairs exhibited distinct separations of particle populations attributed to different rime degrees and particle growth processes. Figure 12 presents a schematic DWR-MDV-polarimetric variable diagram for this case.

Regions that included fallstreaks were dominated by the aggregation process, where the $DWR_{KaW}$ increased with the magnitude of MDV corresponding to the scattering calculations for low- to middle- rime degree aggregate particles (e.g., marked 1 in Fig. 12). The $DWR_{KaW}$ values further increased at lower altitudes of the fallstreaks as reflectivity increased. $Z_{DR}$ and $K_{DP}$ values were 0.6 dB and 0.8 ° $km^{-1}$, respectively. The small $Z_{DR}$ values in the lower region in conjunction with the MDV and Doppler spectrum width measurements suggested further ice growth produced by aggregation. Larger $K_{DP}$ in the fallstreaks represented high number concentration ice particles generated aloft that facilitated aggregation. A possible light riming in a turbulence region could increase the mass of individual particles, hence $K_{DP}$ (e.g., marked 2 in Fig. 12). A sublimation process apparent near the ground at the beginning of precipitation might result in dissipating branches and/or edges of aggregates and decreasing the mean volume diameters. This caused little increase of DWR and kept $K_{DP}$ large.

Characteristics of riming were discerned in other regions where several different particle populations were expected. Associated with a population of lower rime aggregates, $DWR_{KaW}$ increased from near zero to 10 dB while the magnitude of MDV increased from near zero to 0.8 m $s^{-1}$. $K_{DP}$ and $Z_{DR}$ slightly decreased as $DWR_{KaW}$ increased, which were consistent with aggregate particles accompanied by the decrease of their density, increase of their aspect ratios, and more random orientation (e.g., marked 1 in Fig. 12). Another particle population which was expected to have larger degrees of riming was distinguished from the particle populations with smaller degrees of riming using the $DWR_{KaW}$-versus-MDV diagram (e.g., marked 3 in Fig. 12); it had an increase of $DWR_{KaW}$ similar to that for aggregates with lower riming but the magnitude of MDV was around 2–2.5 m $s^{-1}$ (approximately 1–1.5 m $s^{-1}$ larger than that for the former particle population). $K_{DP}$ and $Z_{DR}$ rapidly decreased to near zero when $DWR_{KaW}$ increased suggesting a rapid particle growth. Although $DWR_{KaW}$ also strongly depends on particle shape (in addition to dependence on particle size), the increase of the magnitude of MDV was likely attributed to the ice particle growth. In the lower altitudes, the occurrence of the higher rime degree particle populations increased as the magnitude of MDV reached 3.5 m $s^{-1}$, while $K_{DP}$ and $Z_{DR}$ at a given $DWR_{KaW}$ were smaller as compared to the upper region. These characteristics suggest further riming and increase of aspect ratios. The $DWR_{KaW}$–MDV diagrams also depicted the early stage of riming where $Z_{DR}$ increased while the magnitude of MDV increased collocated with small increases of $DWR_{KaW}$ and $K_{DP}$ (e.g., marked 2 in Fig. 12). The other DWRs (i.e., $DWR_{KKa}$ and $DWR_{KW}$) as a function of MDV as well





as coupling with the polarimetric variables also showed consistent characteristics, indicating that they are very useful to distinguish between riming and aggregation processes.

This study illustrated the capabilities of DWR measurements coupled with MDV and polarimetric measurements to discern riming and aggregation processes, which have been often observed by single-frequency radar measurements but not well separated. This approach will improve quantitative estimations of snow amount (i.e., IWC, snow rate) and microphysical quantities such as rime mass fraction (e.g., Moisseev et al., 2017; Li et al., 2018). Dual-frequency measurements

coupled with MDV -typically available from all cloud radar systems- not only would be more practical than the triple frequency measurements (since they only involve two radars)) but they are more effective in separating the two processes as well. Such systems, when used in synergy with polarimetric observations, common in research and weather networks (e.g. Kollias et al., 2020a, NWS WSR-88D radars) can reveal complex microphysics as presented in this study. Shorter

wavelength radars and lidars as well as microwave radiometers can be complementarily used for better capturing the presence of supercooled liquid droplets and the riming process (e.g., Lamer et al., 2020; Tridon et al., 2020).

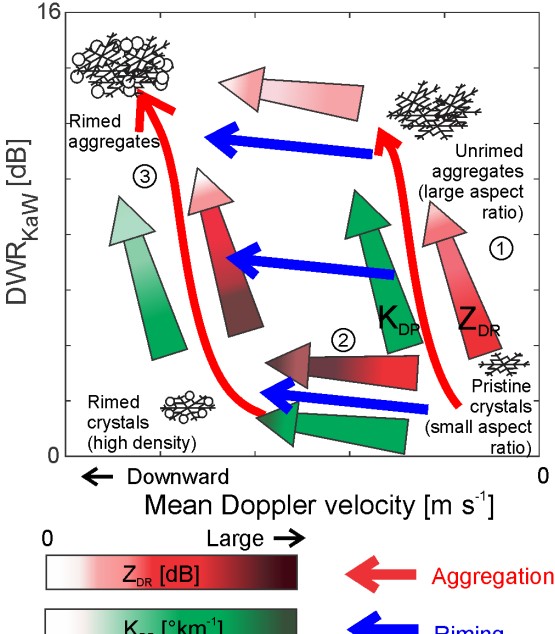

Figure 12: A schematic DWR-MDV-polarimetric variable diagram based on the observation in this study.





Appendix


*Calculations of DWR and mean Doppler velocity for aggregated snowflakes using the self-similar Rayleigh-Gans approximation*

To evaluate the observed DWRs and mean Doppler velocity, we calculated the radar reflectivities
and mean Doppler velocities at the three frequencies (i.e., 24.0 GHz, 35.5 GHz, and 94.0 GHz)
using the radar backscattering cross section database obtained from the self-similar Rayleigh–Gans
approximation (SSRGA) method proposed by Hogan and Westbrook (2014). The SSRGA uses
the Rayleigh-Gans approximation and its extension for an ensemble of particles, for which
horizontal orientation with no canting was employed. The SSRGA employs simple mathematical
formulation which is very efficient in the numerical implementation and produces more realistic
scattering properties compared to spheres/spheroids models, taking into account the internal
structure of aggregates (e.g., Hogan and Westbrook, 2014; Hogan et al., 2017; Tyynelä et al., 2013;
Leinonen et al., 2013; Tridon et al., 2019).

In this work, the SSRGA was adopted to calculate radar backscattering cross sections at a vertical
incident angle for individual aggregate particles with different rime degree (i.e. effective liquid
water path) and size modeled by Leinonen and Szyrmer (2015) and Hogan and Westbrook (2014),
similar to Tridon et al. (2019). Table A1 lists the particle models with different rime degrees used
in the present study. To compute the radar reflectivity from the radar backscatter signals from the
database, we used a gamma distribution as a particle size distribution (PSD), where water-equivalent mass-weighted diameter ($D_m$) varied from 0.1 mm to 2.5 mm with a fixed shape
parameter ($\mu$) of 0 and 4.

Mean Doppler velocity at 1000 hPa was computed for each particle model and each PSD using the
radar backscatter signals and a particle terminal velocity model by Hogan and Westbrook (2014).
For the all MDV presented in this study, negative values represent downward motions.

Table A1: Particle models used in the present study.

| Particle model | Name in figure |
|---|---|
| Leinonen and Szyrmer (2015) unrimed aggregate model (model A) | LS15A0.0kg/m$^2$ |
| Leinonen and Szyrmer (2015) rimed aggregate model (model A) with effective liquid water path of 0.5 kg m$^{-2}$ | LS15A0.5kg/m$^2$ |
| Leinonen and Szyrmer (2015) rimed aggregate model (model A) with effective liquid water path of 2.0 kg m$^{-2}$ | LS15A2.0kg/m$^2$ |
| Leinonen and Szyrmer (2015) rimed aggregate model (model B) with effective liquid water path of 0.5 kg m$^{-2}$ | LS15B0.5kg/m$^2$ |
| Leinonen and Szyrmer (2015) rimed aggregate model (model B) with effective liquid water path of 2.0 kg m$^{-2}$ | LS15B2.0kg/m$^2$ |
| Hogan and Westbrook (2014) | HW14 |


*Data availability.* The SBRO radar data are available at the SBU Academic Commons.



*Author contributions.* Data collection and analysis were made by MO. Conceptualization of the method, interpretation, and writing were shared between MO, PK, SM, AR, and AB. Scattering calculation using SSRGA was made by the AB's group.

*Competing interests.* The authors declare that they have no conflict of interest.

*Acknowledgements.*
M. Oue, P. Kollias, S. Matrosov, and A.V. Ryzhkov were supported by the National Science Foundation grant # 1841246. A. Battaglia was supported by Atmospheric System Research (grant no. DE-SC0017967). We thank Frederić Tridon of University of Cologne for providing the lookup tables of SSRGA scattering properties and Samantha Nebylitsa of University of Miami for processing the MASC and Parsivel data and retrieving particle properties. We also thank Matthew Miller and Sandra Yuter of North Carolina State University for supporting the MASC observations and providing its data.

Financial supports.
This research was supported by the National Science Foundation grant # 1841246.





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
