# Peer review of "Analysis of the Microphysical Properties of Snowfall Using Scanning Polarimetric and Vertically Pointing Multi-Frequency Doppler Radars"

_Atmospheric Measurement Techniques, 2021_

## Referee Comment (RC2)

**Review of Combination Analysis of Multi-Wavelength, Multi-Parameter Radar Measurements for Snowfall**

by Mariko Oue, Pavlos Kollias, Sergey Y. Matrosov,
Alessandro Battaglia, and Alexander V. Ryzhkov

May 3, 2021

**1  Short description**

In this paper, the authors combine observations from vertically pointing radars (K- and W-Band) and polarimetric scanning Ka-band radar to group the ice particles into different classes and growth processes. Ground-based observations from an optical disdrometer (PARSIVEL) and a Multi-Angle Snowflake Camera are used to support the classification and the growth process identification. The authors show that analyses based only on dual-wavelength ratios (DWR) from K, Ka- and W-Band radars do not allow grouping the observation into different ice/snow classes. However, the paper indicates that combining DWR with mean Doppler velocity or differential Doppler velocity makes the classification possible. This last approach allows sites equipped with dual-frequency observations to distinguish between different hydrometeor classes. I, therefore, recommend this paper for publication at AMT, but I ask the author to address the minor issues listed below.

**2  Minor comments**

In the methods section, the authors indicate that the technique proposed by Tridon et al.(2020) to retrieve liquid-water content was applied after adaptations. It would be beneficial for future works if the authors describe the motivation for those adaptations.

**3  Technical suggestion**

The plots in the PDF file suggest that the authors are using the default jet colourmap. This colourmap is known for having a non-uniform transition between the different colours, and it is not friendly for colourblind people. An alternative to the jet colourmap is the newly available turbo colourmap. Below, you can find a link to a brief comparison between turbo and jet colourmaps. Please consider using turbo or some other colourmap that would improve the accessibility of the figures. For the density plots, it would be beneficial to use a perceptually uniform sequential colourmap (see links below).

**Comparison between turbo and jet colourmaps**:
https://ai.googleblog.com/2019/08/turbo-improved-rainbow-colormap-for.html.

**Links for turbo colourmap**:
python: https://matplotlib.org/stable/tutorials/colors/colormaps.html
matlab: https://www.mathworks.com/help/matlab/ref/turbo.html

**Additional colourmap comparison**:
*The misuse of colour in science communication*, https://www.nature.com/articles/s41467-020-19160-7

**Perceptually uniform colourmaps**
python: https://matplotlib.org/stable/tutorials/colors/colormaps.html
Matlab: https://www.mathworks.com/matlabcentral/fileexchange/51986-perceptually-uniform-colormaps

**4 Figure issues**

Figure 1:
The identification from panel d is missing. The hollow circles in cyan and yellow from panels a, c, and d are not easy to distinguish from the white background. What if the authors use filled circles.

Figure 5:
The comparison between the different panels would be easier if they had the same vertical range. Suggestion panels d, e, and f could have the same range from panels a, b, and c.

Figure 8:
The cyan and yellow curves in panel d are difficult to distinguish from the density plot in the background.

Figure 9:
The cyan curve in panels a, d and g is difficult to distinguish from the density plot in the background.

Figure 10:
Panels a and b have a similar issue from figure 9.

**5 Typos**

Pg 7, ln 253: Should it be FMCW instead CFMCW?
Pg 9, ln 333: ... total attenuation **cshould** (typo) then ...

Pg 20, ln 695: ... effects and the **PSDs.** .(there is an extra period) Figure 10 shows dependencies ...

Pg 22, ln 759: These time **seriesare** (missing space) consistent with the ...

---

## Author Comment (AC1)

Responses to Referee #1

Thank you for your appreciation of our study. We have revised the manuscript following the suggestions.

**Major comments:**

- My only major comment is that the paper could benefit from a more specific and meaningful title: this title tells us very little about the measurements taken, the snowfall observed, or the process insights that are made.

Thank you for the important suggestion. We have changed the title to "Analysis of the Microphysical Properties of Snowfall Using Scanning Polarimetric and Vertically Pointing Multi-Frequency Doppler Radars."

Minor comments:

- Figure 1: the key in panel (a) is very difficult to read; it might be possible to add a secondary key showing that solid lines are used for mu=0, and dashed lines for mu=4, so that the number of lines in the first key can be almost halved, and the font size increased. The extents of the axes are almost consistent between panels, but not quite. This would help to make the panel intercomparable.

We have revised Figs 1c and 1d following a coauthor's suggestion; now the new figures use DWR_XW for the X-axis and $DWR_{XKa}$ for the y-axis. The suggestions have been accounted for in the new figure.

- L111: Mason et al. (2018) is the better citation for using Doppler velocity to infer particle properties

Thank you for letting us know about the paper. We have cited the paper in the revised manuscript.

- Titles of sections 2.1, 2.2 and 2.3, and Table 1. This is mostly done well throughout, including the captions to Figures and 1 and 3, but please ensure consistent use of radar bands and frequencies. For example, MRRPro is introduced as a K-band radar in Section 1 (L158) and Section 3 (L306), but not within Section 2.3 or Table 1, whereas both notations are used for KASPR (Section 2.1) and ROGER (Section 2.2). This is all obvious to most readers, but might as well be consistent.

We have clearly mentioned that MRRPro is the K-band (24-GHz) radar in Section 2.

- Figure 2 caption: no Hogan and Westbrook (2017) paper is included in the bibliography; which particle fall velocity model should this refer to?

We have referred to Heymsfield and Westbrook (2010) regarding the particle fall velocity in the revised manuscript and revised the citation.

- Figure 5 & 6: there's a lot of visual comparison asked of the reader in this paper, between different radar variables. Is it possible to use the same height coordinates across different panels for ease of comparison? I'm aware this isn't always practicable, but it is appreciated where possible.

We have used the same height scale in the revised Figures 5 and 6.

- Figures 5 & 11: the time axes here are in decimal hours UTC, which conflicts with the "HH:MM" values referred to throughout the text, and used in Fig. 4f and in the titles to Fig. 6. Again, might as well be consistent, especially where these terms are used to refer to very specific features.

We have converted the decimal hours in Figs. 5 and 11 into the "HH:MM" units.

- L749—50: do I understand this first criterion correctly, that you remove data that relate to particles that are both smaller than 1mm and falling faster than 1.5 m/s? This seems to be the correct reading, but on first glance I thought it meant you were excluding all data with particles smaller than 1mm, and all data with velocities greater than 1.5 m/s. A slight rewording might help make this clearer.

We have rephrased the sentences to read: "we removed apparently unrealistic velocity values exceeding 1.5 m s$^{-1}$ associated with particles having diameter less than 1 mm in agreement with Locatelli and Hobbs (1974),"

- Figure 12, L820--854: Is it possible to strengthen the links between this diagram and the features identified in the case study by relating the different stages (1—4 in the diagram) to the different regions in the case study (A—D)? When the paragraph starts "Regions that included fallstreaks were dominated by..." , the reader will want to be reminded of which regions, and in which figures these features were evident. This diagram helps to make more explicit the processes linking regions A to B and C to D, so it's worth using the regions that have been used throughout the rest of the paper.

Thank you for a valuable suggestion. We have added the regions (A-D) in correspondence with the processes (1-3) in Fig. 12 to the paragraph.

Typos:

- L279: a missing space

Done.

- L302: "non-precipitating cloud case"?

Done.

- L333: "should"

Done.

- Table 1: Are these really the range of velocity resolutions used for MRRPro in this study?

0.19 m s$^{-1}$ is the right value used for this study. We have added this to the table.

- L511-2: "...each of which had similar..." I think "similar" is ambiguous here, since it could be read to mean that the four regions had similar values to each other, not similar values within each region. Perhaps something like "distinct" or "characteristic" would be better.

We rephrased this to read "similar characteristics of DWRs, MDV, and polarimetry with the region."

- L664: "winter storms"

Done.

- L695: A leftover period.

The extra period has been removed.

- L715: "envelope"

Done.

- Figure 10 caption: I think the reference to Sect 3.4 should now be Appendix A

Thank you for pointing this out. We have revised it.

- L759: "...series are..."

Done.

- L775: "...2D particle projections..."

Done.

- L783—5: "...which heavily rimed having faster fall speeds" is confusing, and either "polarimetric variables" or "polarimetric observations" probably works better.

We rephrased the sentence to read "The ground-based characteristics suggest that the snowflakes were heavily-rimed, larger aggregates, consistent with the observed characteristics of the radar MDV, DWR, and polarimetric observables."

- L800: A missing sentence, or an extra period.

I added a comma after "effective."

- Figure 12: There are no values on the x-axis

Revised.

---

## Author Comment (AC2)

Responses to Referee #2

Thank you for your appreciation of our study. We have revised the manuscript following your suggestions. Taking into account the Referee#1's comment, we have changed the title to "Analysis of the Microphysical Properties of Snowfall Using Scanning Polarimetric and Vertically Pointing Multi-Frequency Doppler Radars."

**Minor comments**

- In the methods section, the authors indicate that the technique proposed by Tridon et al.(2020) to retrieve liquid-water content was applied after adaptations. It would be beneficial for future works if the authors describe the motivation for those adaptations.

We selected the values of the parameters of this technique very close to Tridon et al. (2020), but the parameter settings could depend on radar data and cases. The first adaptation was the gates for averaging of DWR. This adaptation was due to the difference in range resolution of the data. The range resolution of our data was 15 m, so we selected 450 m for the averaging (the original was 500 m). The second adaptation was the window size for the calculation of a variance. Because our DWR data were still noisy after averaging, we used the larger window size, 450 m (the original one was 150 m). We have added this to Sect. 3.1.

**Technical suggestion**

- The plots in the PDF file suggest that the authors are using the default jet colourmap. This colourmap is known for having a non-uniform transition between the different colours, and it is not friendly for colour blind people. An alternative to the jet colourmap is the newly available turbo colourmap. Below, you can find a link to a brief comparison between turbo and jet colourmaps. Please consider using turbo or some other colourmap that would improve the accessibility of the figures. For the density plots, it would be beneficial to use a perceptually uniform sequential colourmap (see links below).

Thank you for the valuable suggestion. We have changed the colormaps and used consistent y-axes for Figures 4 and 5.

**Figure issues**

- Figure 1: The identification from panel d is missing. The hollow circles in cyan and yellow from panels a, c, and d are not easy to distinguish from the white background. What if the authors use filled circles.

We have changed the colormap for the diameter scale and line colors in Fig. 1 and Figs. 8-10. We have revised Figs 1c and 1d following a coauthor's suggestion; now the new figures use DWR_XW for the X-axis and DWR_XKa for the y-axis.

- Figure 5: The comparison between the different panels would be easier if they had the same vertical range. Suggestion panels d, e, and f could have the same range from panels a, b, and c.

Done.

- Figure 8: The cyan and yellow curves in panel d are difficult to distinguish from the density plot in the background.
  Figure 9: The cyan curve in panels a, d and g is difficult to distinguish from the density plot in the background.
  Figure 10: Panels a and b have a similar issue from figure 9.

We have changed the colors of the curves in Figures 8-10.

**Typos**
- Pg 7, ln 253: Should it be FMCW instead CFMCW?

The radar system is a Compact Solid-State Frequency Modulated Continuous Wave radar which is also called "C-FMCW" using the first letter of "Compact." We revised the title of the section.

- Pg 9, ln 333: ... total attenuation c should (typo) then ...

Corrected.

- Pg 20, ln 695: ... effects and the PSDs. .(there is an extra period) Figure 10 shows dependencies ...

Corrected.

- Pg 22, ln 759: These time series are (missing space) consistent with

Corrected.